# Computational Insight into the Nature and Strength of the π-Hole Type Chalcogen∙∙∙Chalcogen Interactions in the XO_2_∙∙∙CH_3_YCH_3_ Complexes (X = S, Se, Te; Y = O, S, Se, Te)

**DOI:** 10.3390/ijms242216193

**Published:** 2023-11-10

**Authors:** Fengying Lei, Qingyu Liu, Yeshuang Zhong, Xinai Cui, Jie Yu, Zuquan Hu, Gang Feng, Zhu Zeng, Tao Lu

**Affiliations:** 1School of Basic Medical Sciences/School of Biology and Engineering, Guizhou Medical University, Guiyang 550025, China; lfy2021074@163.com (F.L.); l.qingyu@foxmail.com (Q.L.); zhongyeshuang@foxmail.com (Y.Z.); xinaicui@163.com (X.C.); yujie@gmc.edu.cn (J.Y.); huzuquan@gmc.edu.cn (Z.H.); 2School of Chemistry and Chemical Engineering, Chongqing University, Daxuecheng South Rd. 55, Chongqing 401331, China; fengg@cqu.edu.cn

**Keywords:** chalcogen bonds, π-hole interactions, hydrogen bonds, molecular complexes, theoretical computations

## Abstract

In recent years, the non-covalent interactions between chalcogen centers have aroused substantial research interest because of their potential applications in organocatalysis, materials science, drug design, biological systems, crystal engineering, and molecular recognition. However, studies on π-hole-type chalcogen**∙∙∙**chalcogen interactions are scarcely reported in the literature. Herein, the π-hole-type intermolecular chalcogen**∙∙∙**chalcogen interactions in the model complexes formed between XO_2_ (X = S, Se, Te) and CH_3_YCH_3_ (Y = O, S, Se, Te) were systematically studied by using quantum chemical computations. The model complexes are stabilized via one primary X**∙∙∙**Y chalcogen bond (ChB) and the secondary C−H**∙∙∙**O hydrogen bonds. The binding energies of the studied complexes are in the range of −21.6~−60.4 kJ/mol. The X**∙∙∙**Y distances are significantly smaller than the sum of the van der Waals radii of the corresponding two atoms. The X**∙∙∙**Y ChBs in all the studied complexes except for the SO_2_**∙∙∙**CH_3_OCH_3_ complex are strong in strength and display a partial covalent character revealed by conducting the quantum theory of atoms in molecules (QTAIM), a non-covalent interaction plot (NCIplot), and natural bond orbital (NBO) analyses. The symmetry-adapted perturbation theory (SAPT) analysis discloses that the X**∙∙∙**Y ChBs are primarily dominated by the electrostatic component.

## 1. Introduction

A chalcogen bond (ChB) [1,2,3,4] is an attractive non-covalent interaction (NCI) between an electron-deficient region related to a chalcogen atom (mainly S, Se, and Te) as Lewis acids and any electron-rich region (lone pairs, π-electrons, anions) as Lewis bases. These electron-deficient regions corresponding to the positive electrostatic potentials can be divided into two categories: σ-holes and π-holes [5,6,7,8,9]. The former is generally located along the extension of the covalent σ-bond involving a chalcogen atom, while the latter is generally located perpendicular to the σ-framework of the molecular entity. Both σ-holes and π-holes are capable of interacting attractively with Lewis bases, and the formed NCIs are called the σ-hole interactions and π-hole interactions [6,7,10,11,12,13], respectively. Previous studies disclose that the attractive nature of such interactions is mainly composed of electrostatic, dispersion, and charge transfer interactions [1,14]. In analogy to the halogen bonds (HaBs) [15,16] which involve group VII element atoms (mainly Cl, Br, and I) as Lewis acid sites, the ChBs exhibit a strong directional nature as a result of the existence of the σ-hole. In addition, the ChBs also exhibit comparable strength to that of the HaBs or hydrogen bonds (HBs) [17] and in some cases even exceed that of the HBs [18,19]. Additionally, it should be noted that the chalcogen atoms can serve not only as the ChB donors owing to the existence of an σ-hole [20,21,22,23] or π-hole [24,25,26] on the chalcogen atoms, but also as the ChB acceptors thanks to the presence of lone pairs of electrons on the chalcogen atoms as in chalcoethers [20,26].

Among the various ChBs, the non-covalent chalcogen**∙∙∙**chalcogen interactions have received substantial research interest in recent years because of their potential applications in organocatalysis [27,28,29,30], materials science [31], drug design [32], biological systems [33,34,35,36], crystal engineering [37,38,39], and molecular recognition [40,41]. Experimental and theoretical studies concerning such interactions have suggested that both the chalcogen atom size and the substituents covalently attached to the chalcogen atom can affect the strength of the ChBs [37,42,43]. Specifically, the ChB becomes stronger in strength as the ChB donor atom increases in size and progressively becomes weaker in strength as the ChB acceptor atom increases in size. Moreover, the strength of the ChB becomes greater as the electron-withdrawing capacity of the substituent covalently linked to the ChB donor atom and the electron-donating capacity of the substituent adjoined with the ChB acceptor atom increase. Gleiter and colleagues [42,43] theoretically studied the binary complexes of CH_3_YCH_3_ and CH_3_YZ (Y = S, Se, Te; Z = CH_3_, CN), and symmetry-adapted perturbation theory (SAPT) analysis demonstrated that dispersion and induction forces are responsible for the formation of chalcogen**∙∙∙**chalcogen interactions. Additionally, statistical analyses of crystal structure surveys have also shown that there is a great number of chalcogen**∙∙∙**chalcogen interactions in small molecules, nucleic acids, proteins, and protein–ligand complexes [33,35,36,44,45,46,47], further suggesting the importance of such chalcogen**∙∙∙**chalcogen interactions.

Currently, the vast majority of investigations concerning ChBs focus on the divalent chalcogen atoms as the ChB donors [18,27,28,29,30,31,32,33,35,36,37,38,39,40,41,42,43,44,45,46,47,48,49,50]. However, the chalcogen atoms can also frequently participate in hypervalent bonding such as tetravalent bonding [21,51,52,53]. Taking the S atom as an example, it behaves as a tetravalent bond in both SF_4_ and SO_2_. The former acts as a ChB donor to form the σ-hole interactions with Lewis bases [19,54], while the latter acts as a ChB donor to form the π-hole interactions with Lewis bases [24,26,55,56]. Scheiner and coworkers theoretically studied the heterodimers of SF_4_ and nitrogen-containing Lewis bases and revealed that the S**∙∙∙**N ChB are stronger in strength than that of a classical hydrogen bond [19]. The same group also computationally studied the S**∙∙∙**O ChBs between SO_2_ and a series of carbonyl-containing molecules, and the results show that the most stable isomer of each complex is mainly stabilized by one S**∙∙∙**O ChB as the primary interaction, supplemented by weak C−H**∙∙∙**O HBs [55]. Recently, Feng and coworkers applied microwave spectroscopy and theoretical calculations to investigate the gas-phase binary complex of SO_2_ with cyclohexanol and found that the two moieties within the four detected isomers are connected together via one dominant S**∙∙∙**O ChB and secondary weak C/O−H**∙∙∙**O=S HBs [56]. Microwave spectroscopic investigation on the SO_2_**∙∙∙**CH_3_SCH_3_ complex demonstrated that the S**∙∙∙**S ChB is the primary interaction with the determined distance of 2.947(3) Å [26]. The SAPT analysis results indicate that the electrostatic interactions mainly dominate the attractive nature of both S**∙∙∙**O and S**∙∙∙**S ChBs in the two above-mentioned complexes [26,56].

To sum up, the investigations on chalcogen**∙∙∙**chalcogen interactions so far mainly correspond to σ-hole interactions. However, reports on π-hole-type chalcogen**∙∙∙**chalcogen interactions are still very scarce [25,26,55,56]. In addition, as far as we know, no studies have been reported on such interactions containing the heavy chalcogen atom Te as a ChB donor or ChB acceptor. Herein, we used quantum chemical calculations to systematically investigate the strength and nature of the π-hole-type intermolecular chalcogen**∙∙∙**chalcogen interactions in the model complexes between XO_2_ (X = S, Se, Te) and CH_3_YCH_3_ (Y = O, S, Se, Te), which was frequently used as a nucleophile to engage in various NCIs like hydrogen bonds [57,58], chalcogen bonds [20,26], and pnictogen bonds [59]. Additionally, the molecular electrostatic surface potential (MESP) [60], quantum theory of atoms in molecules (QTAIM) [61], non-covalent interaction plot (NCIplot) [62], natural bond orbital (NBO) [63], and symmetry-adapted perturbation theory (SAPT) [64] analyses were also conducted to gain a deeper understanding of the π-hole-type chalcogen**∙∙∙**chalcogen interactions. Simultaneously, we have also compared the strength and nature of such interactions with those of the σ-hole-type chalcogen**∙∙∙**chalcogen interactions within the CH_3_YCH_3_**∙∙∙**CH_3_YZ (Y = S, Se, Te; Z = CH_3_, CN) complexes [42,43].

## 2. Results and Discussion

### 2.1. Molecular Electrostatic Surface Potential (MESP)

For the sake of identifying the potential interaction sites in each monomer, the MESPs of the XO_2_ (X = S, Se, Te) and CH_3_YCH_3_ (Y = O, S, Se, Te) monomers were computed at the MP2/aug-cc-pVTZ(PP) level of theory. The MESP maps of these monomers are shown in Figure 1. For all the monomers, the positive potential corresponds to the red-colored region and the negative potential corresponds to the blue-colored region. For the XO_2_ (X = S, Se, Te) monomers, the positive potential regions (π-holes) are located above the chalcogen atoms on both sides of the monomeric plane, exhibiting a perpendicular orientation to the molecular plane. The negative potential regions are located at the surfaces of two O atoms. The most positive electrostatic potential values (*V*_S,max_) are 183.0, 217.8, and 257.7 kJ/mol for the SO_2_, SeO_2_, and TeO_2_ molecules, respectively. The *V*_S,max_ values related to the chalcogen atom become more positive as the chalcogen atom increases in size. This can be primarily attributed to the fact that the polarizability of the X atom becomes larger, and its electronegativity becomes smaller as the size of the chalcogen atom increases. Conversely, the negative potential regions are distributed around the chalcogen atoms for the CH_3_YCH_3_ (Y = O, S, Se, Te) monomers, and the positive potential regions are located on the H atoms of each CH_3_ group. The absolute value of the *V*_S,min_ associated with the chalcogen atom decreases as the chalcogen atomic radius increases. The most negative potential values (*V*_S,min_) are −143.9, −103.6, −99.2, and −91.4 kJ/mol for the CH_3_OCH_3_, CH_3_SCH_3_, CH_3_SeCH_3_, and CH_3_TeCH_3_ molecules, respectively. Accordingly, we can predict that the intermolecular π-hole-type chalcogen**∙∙∙**chalcogen interactions between the X atom of XO_2_ and the Y atom of CH_3_YCH_3_ and the C−H**∙∙∙**O interactions between the O atoms of XO_2_ and the H atoms of CH_3_YCH_3_ can be formed.

### 2.2. Geometrical Structures and Binding Energies of the Studied Complexes

Based on the interaction sites identified by the MESP analysis, the initial structures of the studied 12 complexes were obtained by changing the relative positions between XO_2_ and CH_3_YCH_3_ monomers. Figure 2 shows the geometrically optimized structures of these 12 complexes, and the corresponding Cartesian coordinates for each complex are provided in Appendix A. The binding energies (*E*_B_) and the key geometrical parameters associated with the ChBs within these complexes are summarized in Table 1. One can clearly find that the complexes of CH_3_OCH_3_ with XO_2_ (X = S, Se, Te) possess *C*_1_ symmetry and the remaining complexes have *C*_s_ symmetry. Interestingly, it should be pointed out that the structures possessing *C*_s_ symmetry for the XO_2_**∙∙∙**CH_3_OCH_3_ (X = S, Se, Te) complexes are unstable, with imaginary frequencies. The distances (*R*_ChB_) between two interacting chalcogen atoms in all these complexes vary from 2.603 Å to 3.210 Å (Table 1), which are obviously smaller than the sum (*R*_sum,1_) of van der Waals radii of the corresponding two atoms by 18.6% to 29.4% [65], thus suggesting the formation of a strong π-hole-type chalcogen**∙∙∙**chalcogen interaction. However, it should be also noted that these distances are bigger than the sum (*R*_sum,2_) of covalent radii of the two interacting chalcogen atoms by 18.0% to 56.8% [66]. The π-hole-type X**∙∙∙**Y distances in this work are all significantly shorter than the σ-hole-type X**∙∙∙**Y distances in CH_3_XCN**∙∙∙**CH_3_YCH_3_ (X = S, Se, Te; Y = O, S, Se, Te) complexes [42,43], suggesting that the π-hole-type X**∙∙∙**Y ChBs are stronger in strength than those of the σ-hole-type X**∙∙∙**Y ChBs. In addition, Obenchain and collaborators investigated the π-hole chalcogen-bonded complex formed between SO_2_ and CH_3_SCH_3_ using high-resolution microwave spectroscopy in the gas phase and experimentally determined the intermolecular S**∙∙∙**S distance of 2.947(3) Å [26]. This is in good accordance with the computed theoretical value of 2.920 Å, demonstrating the reliability of the theoretical method utilized in this paper. We also calculated the ratio (λ) between *R*_ChB_ and *R*_sum,1_ to qualitatively evaluate the strength of the π-hole-type chalcogen**∙∙∙**chalcogen interactions. The calculated λ values range from 0.71 to 0.81 for all the studied complexes, further indicating that the relatively strong π-hole-type chalcogen**∙∙∙**chalcogen interactions are formed in the studied complexes. The O**∙∙∙**H distances between one of the O atoms of XO_2_ and one of the H atoms of CH_3_YCH_3_ are determined to be in the range of 2.021–2.704 Å (see Appendix A). These O**∙∙∙**H distances in all the studied complexes except for the SO_2_**∙∙∙**CH_3_OCH_3_ complex are significantly lower than the sum (2.62 Å) of van der Waals radii of the corresponding two atoms. The ∠C−H**∙∙∙**O angle has been determined to be in the range of 102.4–151.8° (see Appendix A). It is important to point out that the O**∙∙∙**H distance gradually decreases and the ∠C−H**∙∙∙**O angle becomes bigger as the Y atomic radius increases for a given XO_2_ (X = S, Se, or Te) subunit. This suggests that the strength of the C−H**∙∙∙**O HB becomes progressively stronger as the size of the Y atom increases.

One can see from Table 1 that the *E*_B_ values of the studied complexes vary from −21.6 kJ/mol for the SO_2_**∙∙∙**CH_3_OCH_3_ complex to −60.4 kJ/mol for the TeO_2_**∙∙∙**CH_3_TeCH_3_ complex. For a given ChB acceptor, the *E*_B_ in absolute value gradually increases with increases in the X atom size, which is very consistent with the above-mentioned MESP analysis results. For the same ChB donor, the *E*_B_ value becomes more negative as the Y atomic radius increases. It is worth noting that there is a good linear correlation between the *E*_B_ and the *V*_S,max_ values (Appendix A) of XO_2_ as well as the *V*_S,min_ values (Appendix A) of CH_3_YCH_3_. In addition, a linear correlation is also found between the *E*_B_ and the X**∙∙∙**Y distance (Appendix A).

It should be also noted that although there exist other isomers involving only hydrogen bonds for the studied model complexes on the potential energy surface, all these hydrogen-bonded isomers are not true minima due to the existence of an imaginary vibrational frequency. This suggests that the π-hole-type chalcogen bonds play a crucial role in stabilizing the studied model complexes. Furthermore, this work mainly focuses on the π-hole-type chalcogen**···**chalcogen interactions; thus, the discussion on these hydrogen-bonded isomers has been omitted for simplicity.

### 2.3. Quantum Theory of Atoms in Molecules (QTAIM) Analysis

For the purpose of estimating the strength and nature of the π-hole-type chalcogen**∙∙∙**chalcogen interactions present in the studied complexes, we performed the QTAIM analysis on the basis of the optimized structures at the MP2/aug-cc-pVTZ(PP) level. One bond critical point (BCP) and bond path (BP) between the two interacting chalcogen atoms in each complex was identified (see Figure 3), demonstrating the formation of the π-hole-type chalcogen**∙∙∙**chalcogen interaction. Interestingly, no BCPs and BPs associated with the C−H**∙∙∙**O HBs were found in the SO_2_**∙∙∙**CH_3_OCH_3_ complex, and there only exists one BCP and BP associated with one C−H**∙∙∙**O HB in the SeO_2_**∙∙∙**CH_3_OCH_3_ complex. However, for all the other complexes, two BCPs and BPs related to two C−H**∙∙∙**O HBs between each O atom of XO_2_ and one H atom of each CH_3_ group of CH_3_YCH_3_ were identified. Table 2 lists the calculated topological parameters at the BCPs including the electron density (*ρ*(r)), Laplacian of electron density (∇^2^*ρ*(r)), and total energy density (*H*(r)). The absolute ratio between local kinetic energy density (*G*(r)) and local potential energy density (*V*(r)) was also computed and is given in Table 2. One can see that the *ρ*(r) value varies from 0.0290 a.u for the SO_2_**∙∙∙**CH_3_OCH_3_ complex to 0.0438 a.u for the TeO_2_**∙∙∙**CH_3_OCH_3_ complex, and the corresponding ∇^2^*ρ*(r) value varies between 0.0184 and 0.0911 a.u, suggesting the formation of a relatively strong interaction between two chalcogen atoms. In addition, the *ρ*(r) value gradually decreases in the order of Y = O > S > Se > Te for the same Lewis acids (XO_2_). Similarly, for the same Lewis bases (CH_3_YCH_3_), the *ρ*(r) value gradually becomes larger as the X atom increases in size. This is in good accordance with the above-mentioned MESP analysis results. The QTAIM analysis results reveal that the π-hole-type chalcogen**∙∙∙**chalcogen interactions in all the studied complexes are moderate strong closed-shell interactions owing to the positive ∇^2^*ρ*(r) values. Apart from the SO_2_**∙∙∙**CH_3_OCH_3_ complex, the *H*(r) values are all negative and the |*G*(r)/*V*(r)| values are less than 1, demonstrating that the natures of these chalcogen**∙∙∙**chalcogen interactions have partial covalent characters.

### 2.4. Non-Covalent Interaction Plot (NCIplot) Analysis

The intermolecular interactions between XO_2_ and CH_3_YCH_3_ were also characterized and visualized by performing the NCIplot analysis, which is based on the electron density and its derivatives. The NCIplot analysis results are graphically displayed in Figure 4, where the weak and strong attractive interactions are represented in green-colored and blue-colored regions, respectively, and the repulsive interactions correspond to the red-colored regions. One can see that one dark blue isofurface between the X atom and Y atom in each studied complex was found, indicating the existence of a strong attractive chalcogen**∙∙∙**chalcogen interaction. Additionally, two green(-blush) isosurfaces between the O and H atoms were also found in all the studied complexes, suggesting the presence of two weak attractive C−H**∙∙∙**O HBs. It should be noted that the attractive HBs in the XO_2_**∙∙∙**CH_3_OCH_3_ (X = S, Se, Te) complexes are the weakest among these HBs. Figure 4 also displays the scatter plots of the electronic reduced density gradient (RDG) vs. the sign(λ^2^)*ρ* for the twelve studied complexes. It is seen that the sign(λ^2^)*ρ* values associated with the X**∙∙∙**Y ChBs and the C−H**∙∙∙**O HBs are all negative, further confirming the presence of intermolecular attractive interactions. However, the sign(λ^2^)*ρ* values for the X**∙∙∙**Y ChBs are significantly more negative than those for the C−H**∙∙∙**O HBs, indicating the strengths of the X**∙∙∙**Y ChBs are obviously stronger than those of the C−H**∙∙∙**O HBs in all the studied complexes. Furthermore, it is worth mentioning that for the given ChB acceptor, the sign(λ^2^)*ρ* value becomes more negative upon going from the S to Te, demonstrating that the X**∙∙∙**Y ChBs are stronger in strength in the order of X = S < Se < Te. This matches very well with the MESP and QTAIM analysis results.

### 2.5. Natural Bond Orbital (NBO) Analysis

To further understand the nature of the X**∙∙∙**Y ChBs in terms of orbital interactions and the corresponding second-order perturbation energy (*E*^(2)^), which can qualitatively represent the strength of the X**∙∙∙**Y ChBs, NBO analysis has been implemented for the studied complexes. Table 3 summarizes the obtained *E*^(2)^ values for orbital interactions associated with the X**∙∙∙**Y ChBs. The results disclose that the interactions between the lone pair (LP) of the Y atoms of CH_3_YCH_3_ and π*(O=X, X = S, Se, Te) antibonding orbital of XO_2_ are the largest contribution to the stabilization of these twelve complexes. One can note that the *E*^(2)^ values for all the LP(Y)→π*(O=X) orbital interactions except for the LP(O)→π*(O=S) orbital interaction in the SO_2_**∙∙∙**CH_3_OCH_3_ complex are relatively large. Indeed, the X**∙∙∙**Y ChB is so strong that the NBO judges it to be a covalent bond in the SeO_2_**∙∙∙**CH_3_TeCH_3_, TeO_2_**∙∙∙**CH_3_SeCH_3_, and TeO_2_**∙∙∙**CH_3_TeCH_3_ complexes. In other words, the NBO considers these three complexes as one single molecular entity. For the same ChB donor, e.g., SO_2_, the *E*^(2)^ value becomes larger as the Y atom becomes bigger in size. Similarly, for the same ChB acceptor, e.g., CH_3_OCH_3_, the *E*^(2)^ value increases upon going from the S atom to the Te atom. The changing trend of the *E*^(2)^ values is in qualitative accordance with that of the *E*_B_ values and the NCIplot findings described above. In addition, Figure 5 also graphically displays the NBO analysis results for the three selected representative complexes. One can clearly see that there exists a large overlap between the LP orbital of the S atom of CH_3_SCH_3_ and the π*(O=X) antibonding orbital of XO_2_ (X = S, Se, Te), suggesting the existence of strong X**∙∙∙**S ChBs.

### 2.6. Symmetry-Adapted Perturbation Theory (SAPT) Analysis

We also carried out the SAPT analysis to gain an in-depth understanding of the intrinsic nature of the studied intermolecular interactions. This method can decompose the total interaction energies (*E*_total_) of the studied complexes into the three attractive components including electrostatics (*E*_elec_), induction (*E*_ind_), and dispersion (*E*_disp_) interactions and one repulsive component of exchange–repulsion (*E*_ex-re_) interaction. Table 4 collects the resulting energetical values for each component. It is evident that the electrostatic component is the largest contributor for the attraction of these interactions, which stabilize the studied complexes. The contribution of this component to the total attractive interaction energy varies between 41% and 51%. The dispersion component is dominant over the induction component in the SO_2_**∙∙∙**CH_3_OCH_3_ complex, whilst the induction component is superior to the dispersion component for all the remaining complexes. It is estimated that these two components account for about 49–59% of the total attractive interaction energies. Conversely, the σ-hole-type chalcogen**∙∙∙**chalcogen interactions are mainly dominated by induction and dispersion components in the CH_3_XCN**∙∙∙**CH_3_YCH_3_ (X = S, Se, Te; Y = O, S, Se, Te) complexes [42,43]. The results of the SPAT analysis also indicate that the total interaction energies range from −31.3 kJ/mol for the SO_2_**∙∙∙**CH_3_OCH_3_ complex to −129.8 kJ/mol for the TeO_2_**∙∙∙**CH_3_TeCH_3_ complex. For the same Lewis acid, the *E*_total_ value becomes more negative with increasing Y atom size. For the same Lewis base, the *E*_total_ value also becomes more negative upon moving from X = S to Te atom. In addition, the changing trend of the total interaction energies obtained from the SAPT analysis has a linear correlation with that of the binding energies in Table 1 (Appendix A).

## 3. Computational Methods 

Full geometry optimizations of both monomers and complexes have been carried out via the MP2 method [67] in combination with the aug-cc-pVTZ basis set [68]. The MP2/aug-cc-pVTZ level of theory has been frequently employed to investigate various chalcogen-bonded complexes owing to its accuracy and reliability, which have been demonstrated in the past [24,25,48,49,54]. The pseudopotential aug-cc-pVTZ-PP basis set [69] obtained from the EMSL new Basis Set Exchange (BSE) library [70] was employed for the Te atom to consider the relativistic effects. The same level was also utilized for conducting harmonic vibrational frequency calculations to verify that all the optimized geometrical structures are real minima with no imaginary frequencies. The binding energies (*E*_B_) of the complexes were computed by applying the following equation:*E*_B_ = *E*_AB_ − *E*_A_ − *E*_B_ + BSSE (1)
where *E*_AB_ denotes the energy of the complex, and *E*_A_ and *E*_B_ represent the energy of the isolated optimized monomers. The counterpoise method [71] was utilized to correct the binding energies by removing the basis set superposition error (BSSE). The Gaussian 16 program [72] was used to execute all the computations described above.

The Multiwfn program [73] was employed to compute the molecular electrostatic surface potentials (MESPs) of the monomers on the electron/Bohr^3^ isosurface and the resulting MESP isosurfaces were visualized utilizing the VMD software (version 1.9.3) [74]. The Bader’s QTAIM analysis was performed at the MP2/aug-cc-pVTZ(PP) level of theory via the Multiwfn program to identity bond paths (BPs) and obtain their topological properties including the electron density (*ρ*(r)), Laplacian of electron density (∇^2^*ρ*(r)), local kinetic energy density (*G*(r)), local potential energy density (*V*(r)), and total energy density (*H*(r)) at the bond critical points (BCPs). The Johnson’s NCIplot approach was applied for characterizing the intermolecular interactions occurring in the studied complexes using the Multiwfn program, and the NCIplot analysis results were also visualized with the VMD program. The information on the charge transfer, orbital interactions, and second-order perturbation energy (*E*^(2)^) in the studied complexes was obtained by performing the NBO analysis using the NBO 3.1 module embedded into Gaussian16 program at the B3LYP-D3(BJ)/def2-TZVP level. The SAPT2+3/aug-cc-pVTZ(PP) level [64] was chosen to perform the SAPT analysis in the PSI4 software (version 1.3.2) [75] for quantitatively understanding the nature of the studied intermolecular interactions.

## 4. Conclusions

In summary, we systematically studied the π-hole-type chalcogen**∙∙∙**chalcogen interactions in a series of model complexes of XO_2_ (X = S, Se, Te) with CH_3_YCH_3_ (Y = O, S, Se, Te) using ab initio calculations in conjunction with QTAIM, NCIplot, NBO, and SAPT methodologies in this work. The binding energies range from −21.55 kJ/mol for the SO_2_**∙∙∙**CH_3_OCH_3_ complex to −61.38 kJ/mol for the TeO_2_**∙∙∙**CH_3_TeCH_3_ complex. The X**∙∙∙**Y distance varies between 2.526 Å and 3.210 Å, which is obviously less than the sum of the van der Waals radii of the corresponding two chalcogen atoms. The QTAIM analysis results suggest that all the X**∙∙∙**Y ChBs are closed-shell interactions, and simultaneously, the X**∙∙∙**Y ChBs possess some degree of covalent character in all the studied complexes, except for the SO_2_**∙∙∙**CH_3_OCH_3_ complex. The strengths of the X**∙∙∙**Y ChBs are clearly stronger than those of the C−H**∙∙∙**O HBs, revealed by performing the NCIplot and NBO analyses. In addition, the π-hole-type X**∙∙∙**Y ChBs are stronger in strength than those of the σ-hole-type X**∙∙∙**Y ChBs in the CH_3_XCN**∙∙∙**CH_3_YCH_3_ (X = S, Se, Te; Y = O, S, Se, Te) complexes [42,43]. The findings of the SAPT analysis indicate that electrostatic interactions are the largest contributor to the stabilization of the studied complexes, but the induction and dispersion interactions also play a key role in stabilizing the studied complexes. Hopefully, the findings obtained from this work will prove valuable to the scientific community engaged in crystal engineering, materials science, drug design, organocatalysis, molecular recognition, and biological systems.

## Figures and Tables

**Figure 1 ijms-24-16193-f001:**
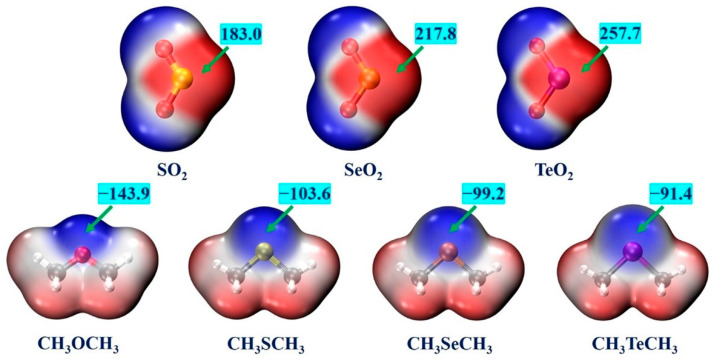
The MESP maps of the XO_2_ (X = S, Se, Te) and CH_3_YCH_3_ (Y = O, S, Se, Te) monomers. The red-colored region indicates the positive potential and the blue-colored region indicates the negative potential. The *V*_S,max_ and *V*_S,min_ values (in kJ/mol) denote the most positive potential and the most negative potential, respectively.

**Figure 2 ijms-24-16193-f002:**
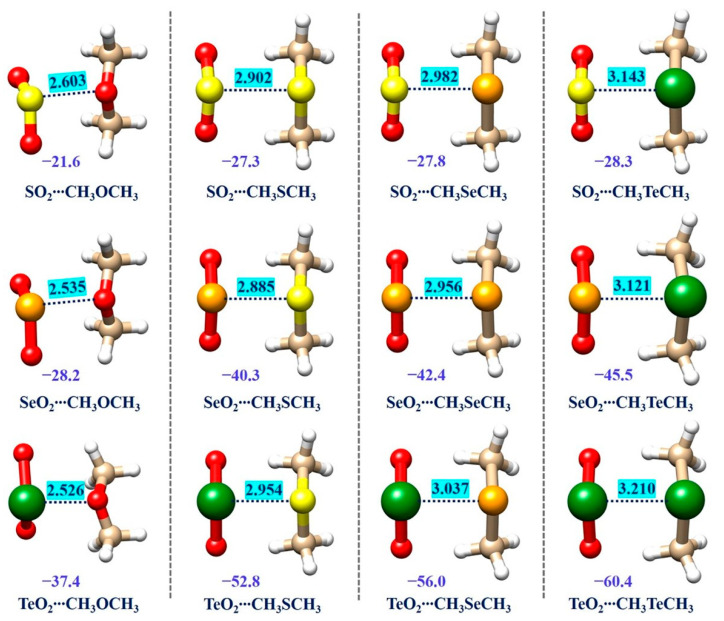
The optimized geometrical structures of the studied complexes. The chalcogen**∙∙∙**chalcogen distances are given in Å. The binding energies in kJ/mol are displayed using blue numbers.

**Figure 3 ijms-24-16193-f003:**
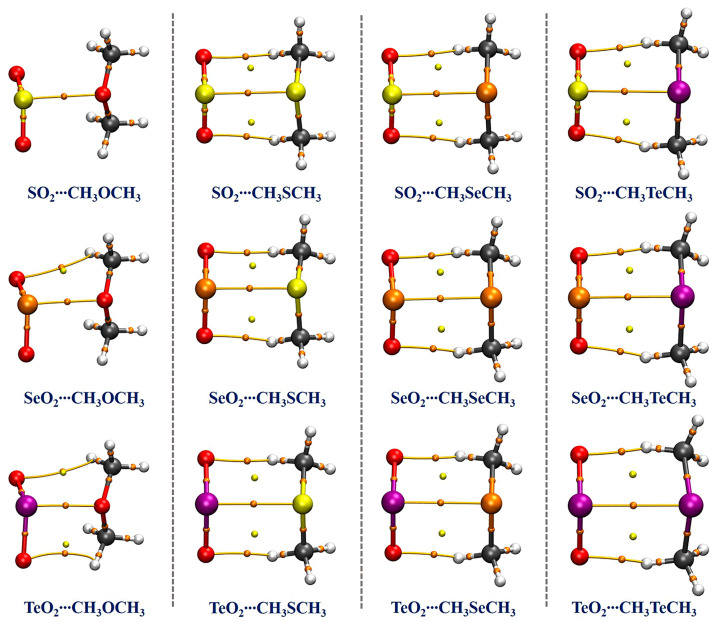
The diagrams of the QTAIM analysis for the studied complexes. The orange and yellow dots represent the bond critical points (BCPs) and ring critical points (RCPs), respectively. The brown lines denote the bond paths.

**Figure 4 ijms-24-16193-f004:**
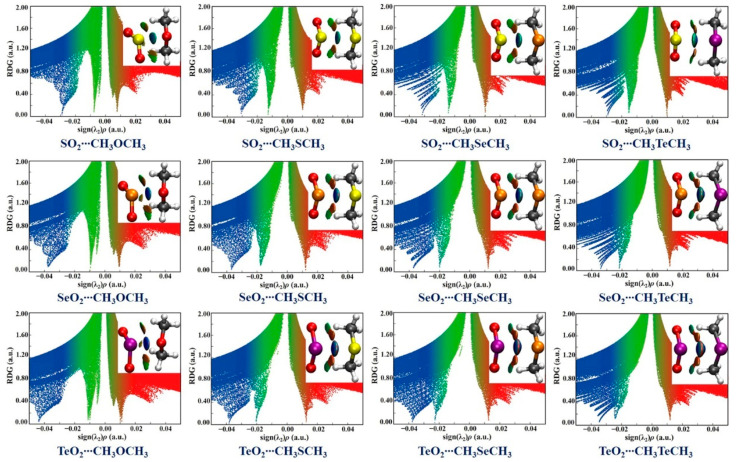
The NCI isosurfaces (*S* = 0.55) and scatter plots of the RDG vs. sign(λ^2^)*ρ* of the twelve studied complexes.

**Figure 5 ijms-24-16193-f005:**
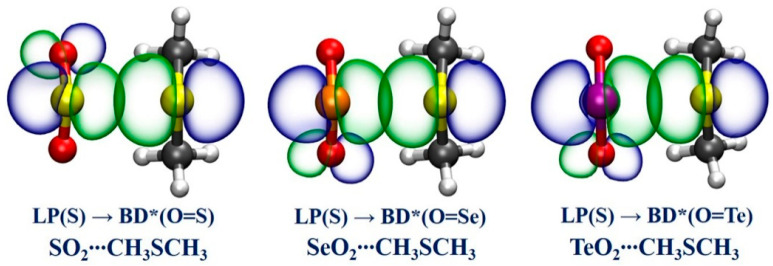
The NBO plots of the donor–acceptor interaction for the three selected representative complexes.

**Table 1 ijms-24-16193-t001:** The binding energies (*E*_B_, kJ/mol) and geometrical parameters associated with the ChBs for the studied complexes.

Complexes	*E* _B_	ChBs	*R*_ChB_ (Å)	*R*_sum,1_ ^a^ (Å)	λ ^b^	*R*_sum,2_ ^c^ (Å)
SO_2_**∙∙∙**CH_3_OCH_3_	−21.6	S**∙∙∙**O	2.603	3.32 (21.6%) ^d^	0.78	1.66 [56.8%] ^e^
SO_2_**∙∙∙**CH_3_SCH_3_	−27.3	S**∙∙∙**S	2.920	3.60 (18.9%)	0.81	2.06 [41.7%]
SO_2_**∙∙∙**CH_3_SeCH_3_	−27.8	S**∙∙∙**Se	2.982	3.70 (19.4%)	0.81	2.19 [36.2%]
SO_2_**∙∙∙**CH_3_TeCH_3_	−28.3	S**∙∙∙**Te	3.143	3.86 (18.6%)	0.81	2.39 [31.5%]
SeO_2_**∙∙∙**CH_3_OCH_3_	−28.2	Se**∙∙∙**O	2.535	3.42 (25.9%)	0.74	1.79 [41.6%]
SeO_2_**∙∙∙**CH_3_SCH_3_	−40.3	Se**∙∙∙**S	2.885	3.70 (22.0%)	0.78	2.19 [31.7%]
SeO_2_**∙∙∙**CH_3_SeCH_3_	−42.4	Se**∙∙∙**Se	2.956	3.80 (22.2%)	0.78	2.32 [27.4%]
SeO_2_**∙∙∙**CH_3_TeCH_3_	−45.5	Se**∙∙∙**Te	3.121	3.96 (21.2%)	0.79	2.52 [23.8%]
TeO_2_**∙∙∙**CH_3_OCH_3_	−37.4	Te**∙∙∙**O	2.526	3.58 (29.4%)	0.71	1.99 [26.9%]
TeO_2_**∙∙∙**CH_3_SCH_3_	−52.8	Te**∙∙∙**S	2.954	3.86 (23.5%)	0.77	2.39 [23.6%]
TeO_2_**∙∙∙**CH_3_SeCH_3_	−56.0	Te**∙∙∙**Se	3.037	3.96 (23.3%)	0.77	2.52 [20.5%]
TeO_2_**∙∙∙**CH_3_TeCH_3_	−60.4	Te**∙∙∙**Te	3.210	4.12 (22.1%)	0.78	2.72 [18.0%]

^a^ *R*_sum,1_ denotes the sum of the van der Waals radii of the corresponding two chalcogen atoms, and the van der Waals radii of the O, S, Se, and Te atoms are 1.52, 1.80, 1.90, and 2.06 Å, respectively [65]. ^b^ λ denotes the ratio of *R*_ChB_ and *R*_sum,1_. ^c^ *R*_sum,2_ denotes the sum of the covalent radii of the corresponding two chalcogen atoms, and the covalent radii of the O, S, Se, and Te atoms are 0.63, 1.03, 1.16, and 1.36 Å, respectively [66]. ^d^ The values in parentheses are the percentage differences between |*R*_ChB_ − *R*_sum,1_| and *R*_sum,1_. ^e^ The values in square brackets are the percentage differences between |*R*_ChB_ − *R*_sum,2_| and *R*_sum,2_.

**Table 2 ijms-24-16193-t002:** Topological properties of the BCPs related to the chalcogen**∙∙∙**chalcogen interactions in all twelve complexes. All the values are reported in a.u.

Complexes	BCP	*ρ*(r)	∇^2^*ρ*(r)	*H*(r)	|*G*(r)/*V*(r)|
SO_2_**∙∙∙**CH_3_OCH_3_	S**∙∙∙**O	0.0290	0.0831	0.0002	1.0087
SO_2_**∙∙∙**CH_3_SCH_3_	S**∙∙∙**S	0.0307	0.0463	−0.0031	0.8266
SO_2_**∙∙∙**CH_3_SeCH_3_	S**∙∙∙**Se	0.0315	0.0395	−0.0037	0.7864
SO_2_**∙∙∙**CH_3_TeCH_3_	S**∙∙∙**Te	0.0292	0.0303	−0.0034	0.7654
SeO_2_**∙∙∙**CH_3_OCH_3_	Se**∙∙∙**O	0.0377	0.0911	−0.0023	0.9172
SeO_2_**∙∙∙**CH_3_SCH_3_	Se**∙∙∙**S	0.0370	0.0420	−0.0053	0.7479
SeO_2_**∙∙∙**CH_3_SeCH_3_	Se**∙∙∙**Se	0.0371	0.0345	−0.0057	0.7153
SeO_2_**∙∙∙**CH_3_TeCH_3_	Se**∙∙∙**Te	0.0340	0.0254	−0.0051	0.6923
TeO_2_**∙∙∙**CH_3_OCH_3_	Te**∙∙∙**O	0.0438	0.0893	−0.0063	0.8202
TeO_2_**∙∙∙**CH_3_SCH_3_	Te**∙∙∙**S	0.0383	0.0328	−0.0069	0.6871
TeO_2_**∙∙∙**CH_3_SeCH_3_	Te**∙∙∙**Se	0.0378	0.0255	−0.0069	0.6576
TeO_2_**∙∙∙**CH_3_TeCH_3_	Te**∙∙∙**Te	0.0346	0.0184	−0.0058	0.6419

**Table 3 ijms-24-16193-t003:** The donor–acceptor orbital interactions and the corresponding *E*^(2)^ value (kJ/mol) related to the X**∙∙∙**Y ChBs in all the studied complexes ^a^.

Complexes	Donor	Acceptor	*E* ^(2)^
SO_2_**∙∙∙**CH_3_OCH_3_	LP(O)	BD*(O=S)	7.7
SO_2_**∙∙∙**CH_3_SCH_3_	LP(S)	BD*(O=S)	55.3
SO_2_**∙∙∙**CH_3_SeCH_3_	LP(Se)	BD*(O=S)	66.2
SO_2_**∙∙∙**CH_3_TeCH_3_	LP(Te)	BD*(O=S)	70.1
SeO_2_**∙∙∙**CH_3_OCH_3_	LP(O)	BD*(O=Se)	44.0
SeO_2_**∙∙∙**CH_3_SCH_3_	LP(S)	BD*(O=Se)	109.3
SeO_2_**∙∙∙**CH_3_SeCH_3_	LP(Se)	BD*(O=Se)	131.5
SeO_2_**∙∙∙**CH_3_TeCH_3_	LP(Te)	BD*(O=Se)	NA ^b^
TeO_2_**∙∙∙**CH_3_OCH_3_	LP(O)	BD*(O=Te)	74.1
TeO_2_**∙∙∙**CH_3_SCH_3_	LP(S)	BD*(O=Te)	137.3
TeO_2_**∙∙∙**CH_3_SeCH_3_	LP(Se)	BD*(O=Te)	NA ^b^
TeO_2_**∙∙∙**CH_3_TeCH_3_	LP(Te)	BD*(O=Te)	NA ^b^

^a^ LP represents lone pair, and BD* represents antibonding orbital. ^b^ The corresponding ChB is so strong that the NBO judges it to be a covalent bond.

**Table 4 ijms-24-16193-t004:** The energetical values of the attractive and repulsion components as well as total interaction energies for the studied complexes obtained by using the SAPT approach at the SAPT2+3/aug-cc-pVTZ(PP) level of calculation ^a^.

Complexes	*E* _elec_	*E* _ind_	*E* _disp_	*E* _ex-re_	*E* _total_
SO_2_**∙∙∙**CH_3_OCH_3_	−58.9(49%) ^b^	−27.5(22%)	−35.0(29%)	90.1	−31.3
SO_2_**∙∙∙**CH_3_SCH_3_	−79.2(43%)	−55.3(30%)	−48.2(27%)	139.6	−43.2
SO_2_**∙∙∙**CH_3_SeCH_3_	−88.6(43%)	−65.4(32%)	−52.8(25%)	160.3	−46.4
SO_2_**∙∙∙**CH_3_TeCH_3_	−87.6(42%)	−65.5(31%)	−55.2(27%)	160.3	−48.0
SeO_2_**∙∙∙**CH_3_OCH_3_	−95.2(49%)	−53.4(27%)	−47.5(24%)	147.2	−48.9
SeO_2_**∙∙∙**CH_3_SCH_3_	−120.3(43%)	−95.8(34%)	−65.5(23%)	206.6	−75.0
SeO_2_**∙∙∙**CH_3_SeCH_3_	−134.4(43%)	−108.5(34%)	−71.0(23%)	230.9	−83.0
SeO_2_**∙∙∙**CH_3_TeCH_3_	−129.8(41%)	−114.4(36%)	−74.4(23%)	228.7	−89.9
TeO_2_**∙∙∙**CH_3_OCH_3_	−145.7(51%)	−80.3(28%)	−59.3(21%)	204.6	−80.7
TeO_2_**∙∙∙**CH_3_SCH_3_	−154.9(46%)	−104.4(31%)	−77.9(23%)	228.7	−108.5
TeO_2_**∙∙∙**CH_3_SeCH_3_	−166.1(45%)	−117.4(32%)	−83.6(23%)	248.2	−119.0
TeO_2_**∙∙∙**CH_3_TeCH_3_	−161.8(44%)	−119.5(32%)	−87.8(24%)	239.2	−129.8

^a^ All the energetical values are given in kJ/mol. ^b^ The parenthesized values indicate the proportion of each attractive component contributing to the total attractive interactions.

## Data Availability

The data that support the findings of this study are available within the article and its Appendix A.

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
