# Peer review of "Computational Insight into the Nature and Strength of the π-Hole Type Chalcogen∙∙∙Chalcogen Interactions in the XO2∙∙∙CH3YCH3 Complexes (X = S, Se, Te; Y = O, S, Se, Te)"

_ijms, 2023, doi:10.3390/ijms242216193_

Round 1

Reviewer 1 Report

Comments and Suggestions for Authors

Review of IJMS-2692285

This article presents detailed computational studies of sigma-hole interactions involving chalcogen atoms, in model complexes.  The work is well motivated and carried out with skill, and the story is clear.  I recommend publication of this article subject to the following minor revisions:

1)    The authors throughout use MP2 method with aug-cc-pVTZ basis set.  Some additional discussion motivating the use of this method would help.  Why were not dispersion corrected DFT methods used instead, for example?  Were other methods used for benchmarking.

2)    While this study naturally is focused on the chalcogen sigma-hole interaction, it is anticipated that even for the smallest clusters examined (e.g., SO2 – dimethyl ether) there are multiple conformational minima on the potential energy surface.  It would help the reader to know how important the sigma-hole complexes are with respect to other types of binding interactions.  This also speaks to the importance of the study. 

3)    It would also be helpful to the reader to contrast the sigma-hole interactions of the chalcogens with those of the halogens, which have been more extensively studied.

Comments on the Quality of English Language

No main issues detected.

Reviewer 2 Report

Comments and Suggestions for Authors

The manuscript of Zeng, Lu, and co-authors describes a nice quantum chemical computational study on π-hole type intermolecular chalcogen∙∙∙chalcogen interactions in the model complexes formed between XO2 (X = S, Se, Te) and CH3YCH3 (Y = O, S, Se, Te). The computational analysis was perfomed by using electrostatic potential, the quantum theory of atoms in molecules (QTAIM), non-covalent interaction plot (NCIplot), natural bond orbital (NBO), and symmetry-adapted perturbation theory (SAPT)  analyses. The study was well designed and conducted, and the manuscript is well-written and clearly presented. On this basis, I recommend the acceptance of this manuscript for publication in the International Journal of Molecular Sciences as it is.

Author Response

The manuscript of Zeng, Lu, and co-authors describes a nice quantum chemical computational study on π-hole type intermolecular chalcogen∙∙∙chalcogen interactions in the model complexes formed between XO2 (X = S, Se, Te) and CH3YCH3 (Y = O, S, Se, Te). The computational analysis was perfomed by using electrostatic potential, the quantum theory of atoms in molecules (QTAIM), non-covalent interaction plot (NCIplot), natural bond orbital (NBO), and symmetry-adapted perturbation theory (SAPT)  analyses. The study was well designed and conducted, and the manuscript is well-written and clearly presented. On this basis, I recommend the acceptance of this manuscript for publication in the International Journal of Molecular Sciences as it is.

Response: We are very grateful to this reviewer for his/her positive comments.